# Development of a Prediction Model to Estimate the Glycemic Load of Ready-to-Eat Meals

**DOI:** 10.3390/foods10112626

**Published:** 2021-10-29

**Authors:** Hosun Lee, Mihyang Um, Kisun Nam, Sang-Jin Chung, Yookyoung Park

**Affiliations:** 1Department of Medical Nutrition, Kyung Hee University, 1732 Deogyeong-daero, Giheung-gu, Yongin-si 17104, Gyeonggi-do, Korea; hhho_o@hanmail.net (H.L.); judy-abbott@hanmail.net (M.U.); 2Corporate Technology Office, Pulmuone Co., Ltd., 280 Gwangpyeong-ro, Seoul 06367, Gangnam-gu, Korea; ksnama@pulmuone.com; 3Department of Foods and Nutrition, Kookmin University, 77 Jeongneung-ro, Seoul 02707, Seongbuk-gu, Korea; schung@kookmin.ac.kr

**Keywords:** GL, glycemic load, ready-to-eat (RTE) meal, prediction formula, GI, glycemic index

## Abstract

The glycemic index (GI) and glycemic load (GL) of a single food item has been used to monitor blood glucose level. However, concerns regarding the clinical relevance of the GI or GL have been raised on their applicability to a combination of several foods consumed as meal. This study aimed to investigate the glycemic response after consuming commercially purchased ready-to-eat meal and to develop the GL prediction formula using the composition of nutrients in each meal. Glycemic responses were measured in healthy adults with various mixed meals comprising approximately 25 g, 50 g, and 75 g of carbohydrates. After fasting, participants consumed test meals, and the glycemic response was measured for a subsequent 120 min. The GI and GL values for mixed meals were calculated as area under curve for each participant. For the prediction formula, 70 mixed meals were analyzed, of which the GI and GL values of 64 participants were used. The prediction formula produced was as follows: GL = 19.27 + (0.39 × available carbohydrate) – (0.21 × fat) – (0.01 × protein^2^) – (0.01 × fiber^2^). We hope that this prediction formula can be used as a useful tool to estimate the GL after consuming ready-to-eat meals.

## 1. Introduction

According to the latest data released by the International Diabetes Federation, there were 463 million people (9.3% of the world’s population) with diabetes between the ages of 20 and 79 years worldwide in 2019. This is an increase of 38 million people from the year 2017. Considering this increasing trend, the number of people with diabetes worldwide is expected to reach 578 million (10.2% of the world’s population) by 2030 and 700 million (10.9% of the world’s population) by 2045, the number of people with diabetes worldwide will increase of 51% compared to 2019 [1]. Therefore, in order to prevent the ever-increasing number of diabetic patients, it is time for a prevention strategy plan that can be easily utilized by the general people.

Diabetes has long been considered a disorder of carbohydrate metabolism and chronically elevated blood glucose levels [2]. Thus, a primary goal in the management of diabetes is the regulation of blood glucose to achieve a near-normal blood glucose level. Carbohydrates in foods cause an increase in blood glucose level after ingestion. Usually, the slower the food is digested, the slower the increase in blood glucose level after ingestion. Moreover, after ingesting food that is easily digested and absorbed, blood glucose and insulin secretion increases rapidly [3,4,5]. The value obtained by comparing the speed of increase in blood glucose level after taking each single food with the reference food (sugar or bread) is called the glycemic index (GI). As the GI does not take into account serving size, the glycemic load (GL) is utilized, which is a value that reflects the usual one-time intake to the GI, and it is practically used to complement the disadvantage of the GI value [6].

The GI and GL are most influenced by the amount of carbohydrate contained in the food but are also affected by the processing method of the food, the cooking method, the shape (physical properties) of the food, and the meal to eat together. Depending on the nutritional composition of the meals, the gastric emptying time changes; thus, a wide variety of factors, including the insulin response, affect the glycemic response after ingestion. In general, some studies have reported that foods with high dietary fiber content lower blood glucose levels [7], and protein and lipid contents are known to be major factors on delaying an increase in blood glucose level, thereby affecting the GI and GL indices [8]. Therefore, even if the same single food is consumed, it can be hypothesized that there is a difference in the blood glucose index according to the composition and condition of the food consumed together [9,10]. The GI and GL indices that have been studied and commonly in use so far are mainly focused on a single food item, and the glycemic response to the mixed meal has been studied only as a rough estimation method using the blood GI and the carbohydrate content of each single food [11,12]. However, it has been reported that the method of calculating the amount and ratio of carbohydrates by identifying the types of ingredients constituting the mixed meal one by one is time-consuming and complicated, and it even fails to determine the effects of other nutrients on the blood glucose [13]. Therefore, this study aimed to determine the GL response after ingesting various commercially available ready-to-eat (RTE) meals of mixed meal type and develop the GL prediction formula using the amount of carbohydrate, protein, fat, and dietary fiber in each mixed meal. Using the prediction formulas obtained as a result of this study, it is expected that the expected GL can be roughly estimated only by the contents of nutrients when various mixed meals are consumed in the future. Today, for people who consume a variety of food items in the form of a mixed menu and recently more as an RTE meal, if the GL of mixed meal products can be predicted, it is believed to provide information on food choices that are useful for managing blood glucose level.

## 2. Materials and Methods

### 2.1. Participants and Ethics

Participants were recruited through the campus bulletin board, and this study was conducted at Kyung Hee University and Kookmin University from May to August 2017. Thirty-four healthy participants (17 males and 17 females) aged between 20 and 39 years with normal fasting blood glucose levels and no family history of diabetes were included in the study. Through pre-interviews, participants with chronic diseases or thyroid diseases; participants with a pathophysiologic risk factor, such as a gastrointestinal disorder; and a woman who has been previously diagnosed with gestational diabetes were excluded. Changes in blood glucose levels after the ingestion of sugar solution by oral glucose tolerance test (OGTT) were examined as screening at the first visit after an 8 h overnight fasting. We excluded participants with fasting blood glucose level greater than 100 mg/dL. This study was conducted in compliance with the clinical trial ethical standards based on the Declaration of Helsinki and was approved by Kyung Hee University Institutional Review Board (KHKSIRB-17-005).

### 2.2. Research Proceeding Method

The study participants visited 20 times per person for approximately 8 weeks. They were instructed to maintain normal dietary intake and sleep during the period and to limit their alcohol intake, which was confirmed before each experiment. After 8 h of overnight fasting, sugar solution, boiled white rice, and mixed meal were taken at each visit, and the autologous blood glucose test was performed and recorded at 0, 15, 30, 45, 60, 90, and 120 min. Sugar solution and boiled white rice were used as the standard of glycemic response, and individual glycemic response pattern and digestibility were obtained. After washing hands thoroughly and completely removing the water, fingertip blood was collected with ACCU-CHECK Performa (Roche Diagnostics Inc., Busan, Korea). The participants were seated without speaking or great motion during the measurement to rule out the influence on blood glucose.

The food to be used in this study comprised 25 g, 50 g, and 75 g of carbohydrates, following the protocol of the International Standards Organization [14]. First, sugar was dissolved in 250 mL of water at three concentrations of 25 g, 50 g, and 75 g and then supplied twice. Boiled white rice with carbohydrates corresponding to sugar 25 g, 50 g, and 75 g was tested twice. After completing 12 baseline blood glucose tests, including sugar test six times and boiled white rice test six times, the participants consumed various food containing carbohydrates of the similar amount of sugar/boiled white rice. Thirty-four participants and 32 mixed meals were divided into four groups. Each group comprised eight or nine participants, and these participants received eight mixed meals per group. Each mixed meal was repeatedly consumed by at least eight participants. The selection criteria for mixed meal were first commercial products with the same or similar amount of carbohydrates as sugar/boiled white rice; second, foods containing various combinations of nutrients; and third, RTE meal that was easy to cook to minimize the effect of cooking. Mixed meals were cooked along with the standard recipe. After cooking, the food was served to the participants within 1 min to preserve the food’s temperature. Only minimal amount of water (up to 100 mL) was allowed.

### 2.3. Materials and Reagents

To assess the standard glucose response, we used dextrose, anhydrous (OCI Company Ltd., Seoul, Korea), and the boiled white rice used in this study was commercially RTE purchased from O company. Ready-to-eat products comprising 25 g, 50 g, and 75 g of carbohydrates were distributed to eight or nine participants. Their nutrient contents are listed in Table 1.

### 2.4. Calculation of Glycemic Index (GI), Glycemic Load (GL), and GL Prediction Equation

The GI was calculated as the percentage of GL area up to 120 min after ingestion of each sample divided by the area of increase in sugar blood glucose level by the incremental area under the curve (iAUC) method analyzed by GraphPad prism version 7.0 (GraphPad Software, San Diego, CA, USA):$$\mathrm{Glycemic}\;\mathrm{Index}(\mathrm{GI})=\frac{\mathrm{iAUC}\;\mathrm{after}\;\mathrm{food}\;\mathrm{ingestion}}{\mathrm{iAUC}\;\mathrm{after}\;\mathrm{sugar}\;\mathrm{solution}\;\mathrm{ingestion}}\times 100$$

According to Jones et al., [15], the amount of only available carbohydrate (net carbohydrate) excluding dietary fiber was multiplied to GI and was further divided by 100 to calculate the GL:$$\mathrm{Glycemic}\;\mathrm{Load}\;(\mathrm{GL})=\mathrm{GI}\times \frac{\mathrm{availablee}\;\mathrm{carbohydrate}}{100}$$

To develop the iAUC and GL prediction formula models, the data of our previous study [16] conducted in 2014 were added to this study data for the analysis. A repeated measurement mixed model was used to obtain the iAUC prediction formula of mixed meal. To obtain the GL prediction formula, we estimated the linear association between each iAUC and GL of 2014 and 2017 participants. Subsequently, we developed the GL equation by substituting the previously estimated iAUC equation into the linear formula.

### 2.5. Statistical Analyses

The results were analyzed using the Statistical Package for the Social Sciences version 22.0 program. Continuous variables are presented as mean and standard deviation. A comparison of the change in blood glucose concentration with the change in blood glucose concentration after sugar ingestion for 120 min after ingesting mixed meal/boiled white rice was confirmed for its significance tested by the paired t-test. One-way analysis of variance was used for the figure comparing the mixed meal/boiled white rice/sugar, and Duncan’s post-hoc method was used for the post-verification. The paired t-test was used to compare the mean of the GL areas. The significance level (α) of all statistical analyses was set to 0.05. This study used R program and Statistical Analysis System to calculate the prediction formula.

## 3. Results

### 3.1. Baseline Characteristics of the Participants

The baseline characteristics of the participants are presented in Table 2. The average ages of the male (*n* = 17) and female participants (*n* = 17) were 23.2 ± 1.9 and 23.0 ± 2.3 years, respectively. The average height of the male participants was 173.8 ± 5.8 cm, and it was 163.4 ± 4.1 cm in female participants. The average weight of the male participants was 70.2 ± 9.4 kg, and it was 59.4 ± 11.5 kg in female participants. The body mass index of the male participants was 23.2 ± 2.8 kg/m^2^, and it was 22.2 ± 4.0 kg/m^2^ in female participants. The average body fat of the male participants was 17 ± 7.2%, which was within the normal range, but the average body fat of the female participants was 25.5 ± 9.4%, suggesting mild obesity. The average waist-to-hip ratios (WHRs) of the male and female participants were 0.83 ± 0.05 and 0.84 ± 0.06, respectively. The male and female participants’ WHRs were within the normal range. The average sleeping times of the male and female participants were 6.6 ± 0.9 and 6.4 ± 0.9, respectively. The average fasting blood glucose levels of the male and female participants were 92.4±5.5 mg/dL and 92.7±5.3 mg/dL, respectively. The male and female participants’ blood glucose levels were within the normal range. All variables did not show a significant difference between the male and female participants.

### 3.2. Incremental Area under the Curve (iAUC) of Glucose, GI, and GL after Consuming Representative Mixed Meal Containing Approximately 25 g, 50 g, and 75 g of Carbohydrates

The GL index was not significantly different from the GI index, and the highest GL index was obtained for food #3 (15 ± 3%), food #6 (15 ± 3%), and food #5 (fried pork burdock fried rice, 15 ± 6%), and food #7 had the lowest GL index (4 ± 1%). In the mixed meal containing 50 g of carbohydrates, the GI value of food #16 (multigrain cereal), which had the highest sugar content, was shown to be as high as 70 ± 20%. In contrast, food #17 (three bean salad and quinoa with black sesame dressing) with the highest dietary fiber content showed the lowest GI index (18 ± 5%). The GL was also high at 33 ± 10% for food #16 and lowest (5 ± 1%) for food #17. The highest GI index (78 ± 8%) was found in food #25 containing 75 g of carbohydrates (oat rice, fish cutlet with broccoli cream sauce, stir-fried bamboo shoots and mushroom, pickled radish, stir-fried anchovies with peanuts, bean salad), which is one of the mixed meals. In contrast, the lowest GI value (47 ± 12%) was found to be the same for food #29 (brown rice, tofu and grilled short rib patties, white kimchi, seasoned aster, green leafy vegetables with mustard dressing) (Table 3). Figure 1 shows the changes in blood glucose concentration after ingesting mixed meals for each carbohydrate content.

### 3.3. iAUC and GL Prediction Formula

The prediction model was first formulated using repeated measurement mixed model. And further formulated after excluding the data of high iAUC even with high dietary fiber content (dietary fiber content exceeding 13 g, iAUC of greater than 4000) and theoretically insignificant GI values of less than 10 and greater than 100. The estimated result equation is as follows:$$\sqrt{\mathrm{iAUC}}=37.05+0.35\times \left(\mathrm{available}\;\mathrm{carbohydrate}\right)-0.18\times \;\mathrm{fat}\;-0.01\times {\;\mathrm{protein}}^{2}-0.01\times {\;\mathrm{fiber}}^{2}$$

At this time, the significance probability *p*-values of available carbohydrate, fat, protein^2^ and fiber^2^ were 0.0000, 0.0001, 0.0000, and 0.0324, respectively, which are all statistically significant. To obtain the association between GL and iAUC, after the linear regression model was fitted to estimate the linear association between GL and AUC, a method of predicting GL using the estimated AUC was used:(1)$$\mathrm{GL}=\;-22.07\;+\;1.12\;\times \;\sqrt{\mathrm{iAUC}}$$

After obtaining the association between GL and iAUC as above, the following GL prediction model was obtained by substituting the previously obtained iAUC prediction formula:
$$\mathrm{GL}\;=\;-\;22.07\;+1.12\;\times \;\sqrt{\mathrm{iAUC}}$$
$$\begin{array}{l} =\;-\;22.07+1.12\\ \;\times \;[37.05\;+\;0.35\;\times \;\left(\mathrm{available}\;\mathrm{carbohydrate}\right)-\;0.18\;\times \mathrm{fat}\;\\ \;-\;0.01\;\times {\;\mathrm{protein}}^{2}\;-\;0.01\;\times {\;\mathrm{fiber}}^{2}] \end{array}$$
$$=19.27\;+\;0.39\times \;\left(\mathrm{available}\;\mathrm{carbohydrate}\right)\;-\;0.21\times \;\mathrm{fat}\;-\;0.01\;\times {\;\mathrm{protein}}^{2}\;\;-\;0.01\;\times {\;\mathrm{fiber}}^{2}$$
$$∴\mathrm{GL}\;=19.27\;+\;0.39\;\times \;\left(\mathrm{available}\;\mathrm{carbohydrate}\right)\;-\;0.21\;\times \;\mathrm{fat}\;-\;0.01\;\times {\;\mathrm{protein}}^{2}\;\;-\;0.01\;\times {\;\mathrm{fiber}}^{2}$$

The average value of the actual GL value and the estimated value obtained from the model were plotted for comparison and are shown in Figure 2. The *x*-axis is the estimated value of GL obtained from the model, and the *y*-axis is the mean GL value of each actual mixed meal. Based on the results of regression analysis, the regression model appeared as follows: measured GL = −26.99 + 1.7 × predicted GL. The explanatory power was 73%, with a significance of 0.000, and for every increase of each estimated GL value, the actual GL value was increased by 1.7.

## 4. Discussion

This study aimed at non-diabetic patients to determine the GL response after ingesting various commercially available RTE meals of mixed meal type as compared to the single food item and eventually develop the GL prediction formula using the amount of carbohydrate, protein, fat, and dietary fiber in each mixed meal.

In the food group containing approximately 25 g of carbohydrates, food #3 (chocolate cereal bar with nuts) showed the highest GI and subsequently, food #6 (strawberry smoothie) had a high GI, which had relatively high sugar contents (14 g and 22 g, respectively) In addition, food #6 (strawberry smoothie), which is consumed in a liquid form, is hypothesized to have further affected the GI. This is consistent with the results of previous studies in which a food with a non-solid liquid form has increased surface area and digestive enzymes are easier to access, leading to a higher GI [17]. Stanik et al. reported that when sugar is ingested, monosaccharides are rapidly absorbed into the blood and elevated blood glucose level [18]. In contrast, food #7 (crispy tofu in garlic teriyaki sauce) showed the lowest GI, and the above food had the highest protein and fat content among the other foods. According to Hatonen et al., the addition of fat and carbohydrates to mashed potatoes has been reported to lower the glycemic response [13], which is found to be similar to the above results.

Food #17 (three bean salad and quinoa with black sesame dressing) showed the lowest GI because it had the largest dietary fiber content. Munoz et al. reported that when dietary fiber was added to the same diet rather than the ingestion of carbohydrate alone, the postprandial blood glucose concentration was significantly lower [19], which is similar to the results of this study.

The GI of food #25 (oat rice, fish cutlet with broccoli cream sauce, stir-fried bamboo shoots and mushroom, pickled radish, stir-fried anchovies with peanuts, bean salad) was the highest in the food group containing 75 g of carbohydrates. There was no significant difference in the participants between the area of GL after ingesting 75 g of sugar and the area of GL after ingesting food #25 in the participants, from which it is considered that GI is highly calculated. In contrast, the GI was identically the lowest in food #29 (brown rice, tofu and grilled short rib patties, white kimchi, seasoned aster, green leafy vegetables with mustard dressing) with high fat and high protein contents and food #32 (cold buckwheat noodles with spicy sauce), which was provided in a cold state affecting the GI value. As the starch temperature cools down, the particles rejoin, settle, and age, and this aging process decreases enzyme penetration, which finally decreases digestibility and lowers blood glucose level [20].

In the present study, the participants were both men and women, and considering the possibility of difference in glycemic response between men and women due to the effect of hormones [21], at every visit, we instructed the female participants to refrain from performing the experiment during their menstruation cycle. We found that fasting blood glucose level and OGTT (after ingesting 75 g of sugar, glycemic response test) did not have any significant effect on the glycemic response (data not shown). Therefore, we strongly believe that this study is more advantageous to generalize it to normal population because we included both men and women, different from many studies examining glycemic response using only men.

We calculated the prediction formula model using 70 mixed meals to estimate the GL through the contents of carbohydrates, dietary fiber, fat, and protein in food when the mixed meal is ingested. GL shows a positive correlation with available carbohydrate content in food and shows a negative correlation with fat, protein, and dietary fiber content. It is found that carbohydrates significantly increase GL, and fat, protein, and dietary fiber significantly reduce the GL. Wolever et al. described that 90% of the glycemic response in a mixed meal is due to the effect of carbohydrates [22], which is consistent with the above results of this study. For the results of fat lowering the GL, through previous studies, the combinational ingestion of carbohydrates and fats can slow the gastric emptying and increase the secretion of cholecystokinin, resulting in lower glycemic responses [23]. Moreover, according to Quek’s study, the increase in insulin secretion was associated with a decrease in glycemic response when high protein was added to carbohydrates [24]. Finally, dietary fiber forms a gel due to its cell wall structure characteristics and is slowly absorbed in the gastrointestinal tract. It has been shown that dietary fiber not only prevents the spread of sugar and flattens the glycemic response but also slowly increases the response of insulin and gastric inhibitory polypeptide [25], which supports the results of the prediction formula in this study. According to Pi-sunyer’s study, only water-soluble fibrin, which is not an insoluble fibrin, affects postprandial increase in blood glucose level, on which more studies are needed [20].

The limitations of this study are as follows: (1) we could not take into account the differences of the degree of digestion by each individual; thereby, the individuality of GL response was not taken into account. (2) Since all the participants in this study were Korean, we could not generalize the results to other ethnic population. (3) The GL values may be quite different when evaluating other Ready-to-Eat Meals with extreme nutrient composition.

However, different from previous studies [13,26], which compared a limited number of foods by adding other ingredients to the same amount of carbohydrates, in this study, it is meaningful that we provided frequently consumed 32 foods with various contents of macronutrients, and using the result, we tried to determine the association between the glycemic response and regression equation with nutrients. Our prediction model is by far the first prediction model of a mixed meal. Although further validation is needed, currently, using this equation, by reading the nutrition information on the label, it is possible to roughly estimate its GL value without analyzing the GL value one by one. We strongly believe that it can provide useful information about the glycemic response of each food when participants need to control their blood glucose level in general people.

As a follow-up study, we are in the process of evaluating other ready meals containing ingredients not included in the instant meals evaluated in this study.

## Figures and Tables

**Figure 1 foods-10-02626-f001:**
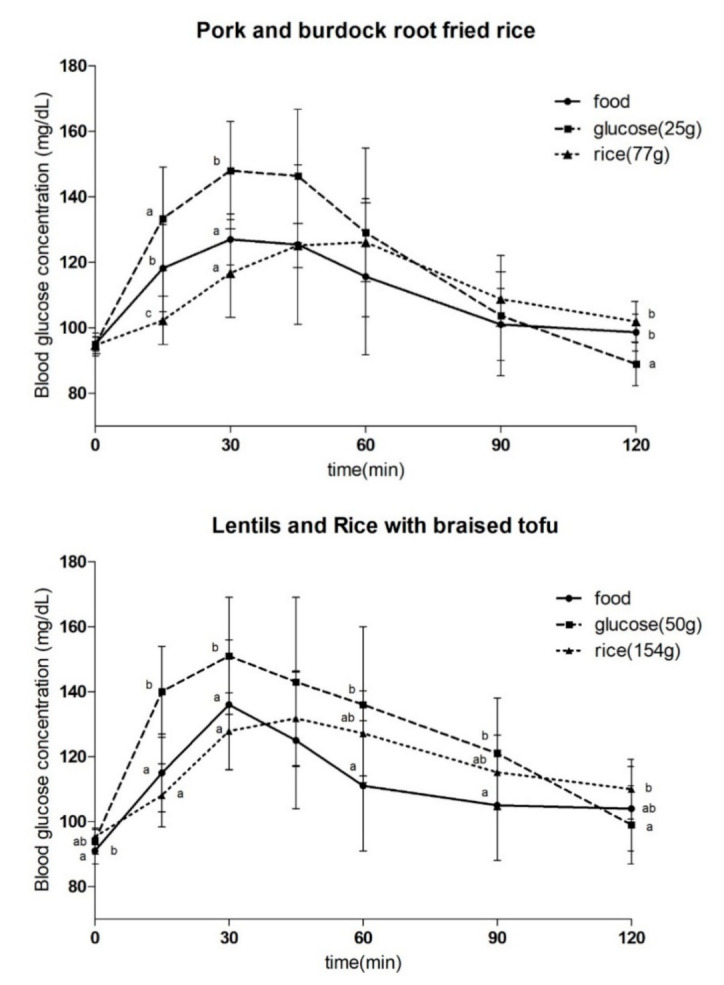
Incremental blood glucose profile for 120 min following the consumption of the selected foods and glucose and rice comprising 25. and 50 g of carbohydrates. Values are presented as mean ± standard deviation. Significant differences by analysis of variance and Duncan’s post-hoc test between glucose levels at certain time points are indicated with different letters (*p* < 0).

**Figure 2 foods-10-02626-f002:**
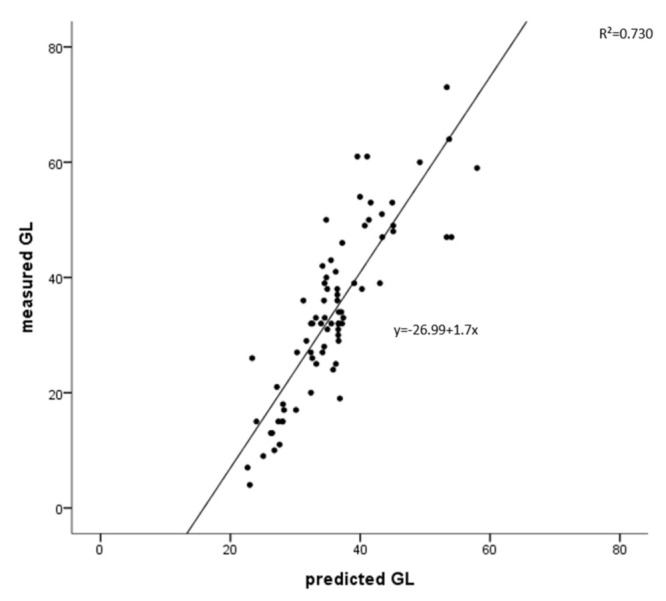
Association between means of measured glycemic load (GL) and predicted GL for 73 dif Figure 2. 0.730, *p* = 0.000).

**Table 1 foods-10-02626-t001:** List of 32 mixed meals based on fixed glucose amount and their ID number.

Mixed Meals Containing Approximately 25 g of Carbohydrates	Mixed Meals Containing Approximately 50 g of Carbohydrates	Mixed Meals Containing Approximately 75 g of Carbohydrates
1. Corn dog	9. Lentils and rice with braised tofu	25. Oat rice, fish cutlet with broccoli cream sauce, stir-fried bamboo shoots and mushroom, pickled radish, stir-fried anchovies with peanuts, bean salad
2. Roasted mixed grain powder	10. Spaghetti with tomato meat sauce and cheese	26. Brown rice, Korean barbecued duck, seasoned sweet potato stalks, dried slices of daikon, stir-fried anchovies with nuts, bean salad
3. Chocolate cereal bar with nuts	11. Black beans and barley rice, spicy braised chicken, bean sprouts, cucumber and balloon flower root salad	27. Multigrain rice with quinoa
4. Sweet potato salad	12. Semi-dried slices of sweet potato	28. Semi-dried slices of sweet potato, corn dog
5. Pork and burdock root fried rice	13. Meat dumplings (10 pieces)	29. Brown rice, tofu and grilled short rib patties, white kimchi, seasoned aster, green leafy vegetables with mustard dressing
6. Strawberry smoothie	14. Korean thistle and barley rice	30. Korean black bean sauce noodles
7. Crispy tofu in garlic teriyaki sauce	15. Corn dogs (two pieces)	31. Brown rice with seasoned aster
8. Sweet pumpkin and chestnut soup	16. Multigrain cereal	32. Cold buckwheat noodles with spicy sauce
	17. Three bean salad (chickpea, pea, and lentil) and quinoa with black sesame dressing	
	18. Korean barbecued beef and mushroom with rice, tomato chili sauce, veggie sticks, braised burdock root	
	19. Chicken breast fried rice with bean sprouts, salad, kimchi, quail egg soy sauce	
	20. Guinea corn and rice, stir-fried octopus, veggie sticks, white kimchi, stir-fried anchovies	
	21. Brown rice, beef stroganoff, grilled vegetables, coleslaw	
	22. Beef and mushroom mixed rice	
	23. Stir-fried rice with seafood and vegetables	
	24. Steamed lotus root rice mixed with sweet pumpkin, mushroom, lentils	

**Table 2 foods-10-02626-t002:** Baseline characteristics of the participants.

Variables	Total (*n* = 34)		
	Male (*n* = 17)	Female (*n* = 17)	*p* ^a^
Age (yrs)	23.2 ± 1.91	23.0 ± 2.3	0.853
Height (cm)	173.8 ± 5.8	163.4 ± 4.1	0.267
Weight (kg)	70.2 ± 9.4	59.4 ±11.5	0.801
BMI (kg/m^2^) ^1^	23.2 ± 2.8	22.2 ± 4.1	0.213
Body fat (%)	17.0 ± 7.2	25.5 ± 9.4	0.282
Skeletal muscle (kg)	31.8 ± 4.9	22.0 ± 2.6	0.020
WHR ^2^	0.83 ± 0.1	0.84 ± 0.1	0.700
Fasting blood glucose (mg/dL)	92.4 ± 5.5	92.7 ± 5.3	0.500

Mean ± standard deviation, ^1^ BMI: body mass index, ^2^ WHR: waist-to-hip ratio. ^a^ *p* values obtained from the independent sample’s *t*-test.

**Table 3 foods-10-02626-t003:** Incremental area under the curve (iAUC) of glucose, glycemic index (GI), and glycemic load (GL) after consuming representative mixed meal containing approximately 25 g and 50 g of carbohydrates.

Food ID		Calories (kcal)	Total Carbohydrate (g)	Dietary Fiber (g)	Sugars (g)	Protein (g)	Fat (g)	Area under the Curve of Glucose (mg * min/dL)		GI (%)	GL
								Glucose	Mixed Meal ^a^		
1	Corn dog	210	26	2	7	5	10	2754 ± 762	1424 ± 401 ***	53 ± 14	13 ± 3
2	Roasted mixed grain powder	125	23	5.5	3	8	1.3	2747 ± 1320	1372 ± 491 ***	54 ± 18	9 ± 3
3	Chocolate cereal bar with nuts	246	25	4	14	8	14	2744 ± 442	1936 ± 272 **	72 ± 14	15 ± 3
4	Sweet potato salad	210	25	1	15	1	12	2877 ± 251	1574 ± 542 **	55 ± 18	13 ± 4
5	Pork and burdock root fried rice	175	26	2	3	11	3	3376 ± 160	1990 ± 129	60 ± 24	15 ± 6
6	Strawberry smoothie	100	25	2	22	1	0	2898 ± 972	2049 ± 684 *	67 ± 13	15 ± 3
7	Crispy tofu in garlic teriyaki sauce	230	26	6	10	18	7	3077 ± 858	499 ± 194 ***	18 ± 70	4 ± 1
8	Sweet pumpkin and chestnut soup	180	25	0	15	4	7	3533 ± 791	1328 ± 399 **	37 ± 6	9 ± 1
9	Lentils and Rice with braised tofu	340	50	5	6	19	8	4139 ± 115	2583 ± 142 **	61 ± 21	27 ± 9
10	Spaghetti with tomato meat sauce and cheese	305	50	2	8	11	7	4075 ± 122	2044 ± 117 ***	49 ± 19	24 ± 9
11	Black beans and barley rice, spicy braised chicken, bean sprouts, cucumber and balloon flower root salad	363	50	7.1	10	23	7.9	4091 ± 1025	2481 ± 883 **	62 ± 19	27 ± 8
12	Semi-dried slices of sweet potato	195	49	5	10	2	0	4200 ± 110	2298 ± 110 **	57 ± 23	25 ± 1
13	Meat Dumplings (10 pieces)	493	52	7	2	22	24	4144 ± 1121	1515 ± 494 ***	40 ± 20	18 ± 9
14	Korean thistle and barley rice	296	50	1	1	6	8	4472 ± 983	2631 ± 910 **	59 ± 15	29 ± 7
15	Corn dogs (2 pieces)	420	52	4	14	10	20	4144 ± 1121	2818 ± 932 **	68 ± 12	33 ± 6
16	Multigrain cereal	240	50	2	18	4	3	4404 ± 915	2959 ± 764 *	70 ± 20	33 ± 1
17	Three bean salad (chickpea, pea and lentil) and quinoa with black sesame dressing	422	46	16	12	14	20	3972 ± 136	576 ± 169 **	18 ± 50	5 ± 1
18	Korean barbecued beef and mushroom with rice, tomato chili sauce, veggie sticks, braised burdock root	305	54	7.1	16	14	3.9	4394 ± 1302	2539 ± 1801	59 ± 19	29 ± 9
19	Chicken breast fried rice with bean sprouts, salad, kimchi, quail egg soy sauce	392	53	6.1	9	19	11.5	3966 ± 1256	2486 ± 400	65 ± 13	31 ± 6
20	Guinea corn and rice, Stir-fried Octopus, veggie sticks, white kimchi, stir-fried anchovies	295	52	6.7	11	14	3.2	3966 ± 1256	2852 ± 539 *	69 ± 11	31 ± 5
21	Brown rice, beef stroganoff, grilled vegetables, coleslaw	320	53	6.4	15	17	4.4	4381 ± 1307	2016 ± 632 *	57 ± 5	27 ± 2
22	Beef and mushroom mixed rice	345	55	4	6	8	11	5373 ± 788	3926 ± 694	74 ± 15	38 ± 8
23	Stir-fried rice with seafood and vegetables	315	54	4	4	11	7	5286 ± 961	3119 ± 494 *	65 ± 9	32 ± 4
24	Steamed lotus root rice mixed with sweet pumpkin, mushroom, lentils	335	58	2	5	7	9	5286 ± 961	2979 ± 1257 *	58 ± 22	32 ± 13
25	Oat rice, fish cutlet with broccoli cream sauce, stir-fried bamboo shoots and mushroom, pickled radish, stir-fried anchovies with peanuts, bean salad	411	71	7	N/A^1^	16	7	6012 ± 147	4619 ± 149	76 ± 80	49 ± 5
26	Brown rice, Korean barbecued duck, seasoned sweet potato stalks, dried slices of daikon, stir-fried anchovies with nuts, bean salad	476	74	7	14	18	12	4585 ± 737	2566±536 **	56 ± 10	38 ± 7
27	Multigrain rice with quinoa	420	74	2	8	9	10	5326 ± 159	3842 ± 992 *	73 ± 12	53 ± 8
28	Semi-dried slices of sweet potato, corn dog	405	75	7	17	7	10	5326 ± 159	2847 ± 116 **	57 ± 25	39 ± 2
29	Brown rice, tofu and grilled short rib patties, white kimchi, seasoned aster, green leafy vegetables with mustard dressing	581	73	11	13	22	22	4646 ± 796	2180 ± 688 **	47 ± 12	29 ± 7
30	Korean black bean sauce noodles	660	99	3	13	23	20	5292 ± 1302	2777 ± 445 *	57 ± 18	54 ± 17
31	Brown rice with seasoned aster	385	68	1	1	8	9	5602 ± 1309	3487 ± 939	65 ± 17	44 ± 11
32	Cold buckwheat noodles with spicy sauce	495	103	0	18	12	4	5602 ± 1309	2621 ± 713 **	47 ± 8	48 ± 8

^a^ *p* values were obtained from the paired t-test. * *p* < 0.01, ** *p* < 0.05, *** *p* < 0.001. N/A = not available.

## Data Availability

The data that support the findings of this study have already been included in the manuscript. Raw data are available from the corresponding author upon reasonable request.

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
