# Peer review of "Development of a Prediction Model to Estimate the Glycemic Load of Ready-to-Eat Meals"

_foods, 2021, doi:10.3390/foods10112626_

Round 1

Reviewer 1 Report

The authors describe a study and algebraic model fit to calculate the GI and GL of multiple food compositions and serving sizes. The study design and gathering of data seems to be sound and it is easily reproducible, but the impact of the study, and specially the context of this work, is lacking.

The main issue this reviewer has with this study is the oversimplification of the goals of the study. The authors design a large study, monitoring ~30 patients over 8 weeks, but only provide a simple algebraic model for the GL computation of an extremely variable and complex process such as meal absorption. There have been huge efforts by the scientific community over the last decade on the characterization of the meal absorption process accounting for meal composition, size, and cooking process, including double and triple trace experiments, to characterize the complete, minute by minute, glucose input into the gut, and posterior absorption by the bloodstream. In contrast, considering the output of the glucose absorption process in this paper, which is only the GL and GI, seems weak in comparison. The authors did not address the relevance of their methods and practical applications when compared with existing literature on the matter.

The authors suggest that this study is geared towards diabetes treatment, but they recruited people without diabetes for this study. Is there a reason for this? it is unclear in the manuscript.

The charts in figure 1 are probable the most interesting part of this manuscript, but they look odd to me, composed in a sort of "collage" without clear reason why. They also look blurry. This reviewer would have liked to see more analysis over this style of data, measuring intra-meal variability, as well as variability between meals.

Why was the time limit of the study 120? Glucose absorption for a mixel meal can trail longer than that, specially for diabetic patients.

In summary, this reviewer has many issues with this manuscript. The authors may reconsider the scope and context of this work, since the experiment they run gathered valuable data, but that data can and must be exploited in a different way to be useful and interesting to the communities it affects.

Author Response

Thank you  for giving us the opportunity to submit a revised draft of the manuscript . We appreciate the time and effort that you dedicated to providing feedback on our manuscript and are grateful for the insightful comments on and valuable improvements to our paper.

we have incorporated most of the suggestions made by the reviewer 1. Those changes are red highlighted within the manuscript. 

Please see below, in attached file for a point -by-point response to the reviewer's comments and concerns.  (files :response to reviewer, revised manuscript)  All page numbers refer to the revised manuscript file with tracked changes.

Reviewer 2 Report

In this manuscript entitled "Development of a Prediction Model to Estimate the Glycemic Load of Ready-to-Eat Meals", the authors examined the glycemic response after consuming commercially purchased ready-to-eat meal and to develop the GL prediction formula using the composition of nutrients in each meal. The authors established prediction formula of GL using collected data. With the increasing number of diabetics and the use of processed foods (ready-to-eat meals), the creation of this formula for predicting GL values is very valuable. I have a few comments, explained below. I hope that my comments are very useful for the improvement of this research.

Comments

(1) Prediction formula The prediction formula produced was as follows: GL = 19.27105 + (0.393566429 × available carbohydrate) (0.205486363 × fat) – (0.006877061 × protein2) – (0.012675566 × fiber2): The number of digits in the coefficient multiplied by each nutrient is too large. It should be reduced. Please consider this point.

(2) L30, L33 in Introduction: What does the 9.3%, 10.2%, and 10.9% represent? Is it the percentage of diabetics in the world population? If so, I think it needs to be explained.

(3) 2.2. Research proceeding method: The time frame (for example, morning, afternoon, or evening) in which the OGTT test was administered is not stated. If there is a relationship between circadian rhythms and elevated blood glucose levels, then the time frame should be stated.

(4) L127: Please show the version of GraphPad prism.

(5) 2.5. Statistical analyses: Authors used the Duncan’s tests as statistical analysis. But, Duncan's test has been pointed out to have problems such as not taking multiplicity. Thus, please change to another multiple tests.

(6) 2.5. Statistical analyses: The details of how you calculated the prediction formula should be described in Materials and Methods section.

(7) Figure 1: Graphs are hard to read. The authors should make a more beautiful chart.

(8) Table 3: Data for all meals should be presented.

(9) Section 3.3: Related to comment 6, the authors should provide details on how you obtained the first equation (L207-208).

(10) Figure 2: Please indicate the units of the axis.

(11) Limitation: In this study, the authors are targeting Korean people and Korean food products. Therefore, the authors do not know if this prediction formula can be used in other countries. I think this should be shown in the limitations of this study.

(12) Limitation: Normally, the GI value of food varies greatly depending on the food. The accuracy of the GL prediction formula using the carbohydrate and dietary fiber values obtained in this study is high. This may be due to the fact that the carbohydrate sources of the foods used in this study were similar ingredients. Therefore, I think there is a possibility that the GL values may be significantly different when evaluating other Ready-to-Eat Meals that contain ingredients that are not included in the Ready-to-Eat Meals evaluated in this study. If this is the case, the limits should be indicated in this regard as well.

Author Response

Thank you for giving us the opportunity to submit a revised draft of the manuscript . We appreciate the time and effort that you dedicated to providing feedback on our manuscript and are grateful for the insightful comments on and valuable improvements to our paper.

we have incorporated most of the suggestions made by the reviewer 2. Those changes are red highlighted within the manuscript.

Please see below, in attached file for a point -by-point response to the reviewer's comments and concerns. (files :response to reviewer, revised manuscript) All page numbers refer to the revised manuscript file with tracked changes.
